# Microendoscopic Posterior Decompression for Treating Thoracic Myelopathy Caused by Ossification of the Ligamentum Flavum: Case Series

**DOI:** 10.3390/medicina56120684

**Published:** 2020-12-10

**Authors:** Satoshi Baba, Ryutaro Shiboi, Jyunichi Yokosuka, Yasushi Oshima, Yuichi Takano, Hiroki Iwai, Hirohiko Inanami, Hisashi Koga

**Affiliations:** 1Department of Orthopaedics, Iwai Orthopaedic Medical Hospital, 8-17-2 Minamikoiwa, Edogawa-ku, Tokyo 133-0056, Japan; r-shiboi@iwai.com (R.S.); j-yokosuka@iwai.com (J.Y.); yoo-tky@umin.ac.jp (Y.O.); luigi.igiul1030@gmail.com (Y.T.); h-iwai@iwai.com (H.I.); inanamihiro@gmail.com (H.I.); hkoga0808@gmail.com (H.K.); 2Department of Orthopaedic Surgery, The University of Tokyo, 57-3-1 Hongo, Bunkyo-ku, Tokyo 113-0033, Japan; 3Department of Spinal Surgery, Japan Community Health Care Organization, Tokyo Shinjuku Medical Center, 5-1 Tsukudo-chou, Shinjuku-ku, Tokyo 162-8643, Japan; 4Department of Orthopaedic Surgery, Ohno Chuo Hospital, 3-20-3 Shimokaizuka, Ichikawa-shi, Chiba 272-0821, Japan

**Keywords:** ossification of ligamentum flavum, thoracic myelopathy, microendoscopic spine surgery, minimally invasive, posterior decompression

## Abstract

*Background and Objectives:* Ossification of the ligamentum flavum (OLF) is a relatively common cause of thoracic myelopathy. Surgical treatment is recommended for patients with myelopathy. Generally, open posterior decompression, with or without fusion, is selected to treat OLF. We performed minimally invasive posterior decompression using a microendoscope and investigated the efficacy of this approach in treating limited type of thoracic OLF. *Materials and Methods:* Microendoscopic posterior decompression was performed for 19 patients (15 men and four women) with thoracic OLF with myelopathy aged between 35 to 81 years (mean age, 61.9 years). Neurological examination and preoperative magnetic resonance imaging (MRI) and computed tomography (CT) were used to identify the location and morphology of OLF. The surgery was performed using a midline approach or a unilateral paramedian approach depending on whether the surgeon used a combination of a tubular retractor and endoscope. The numerical rating scale (NRS) and modified Japanese Orthopedic Association (mJOA) scores were compared pre- and postoperatively. Perioperative complications and the presence of other spine surgeries before and after thoracic OLF surgery were also investigated. *Results:* Four midline and 15 unilateral paramedian approaches were performed. The average operative time per level was 99 min, with minor blood loss. Nine patients had a history of cervical or lumbar spine surgery before or after thoracic spine surgery. The mean pre- and postoperative NRS scores were 6.6 and 5.3, respectively. The mean recovery rate as per the mJOA score was 33.1% (mean follow-up period, 17.8 months), the recovery rates were significantly different between patients who underwent thoracic spine surgery alone (50.5%) and patients who underwent additional spine surgeries (13.7%). Regarding adverse events, one patient experienced dural tear, another experienced postoperative hematoma, and one other underwent reoperation for adjacent thoracic stenosis. *Conclusion:* Microendoscopic posterior decompression was applicable in limited type of thoracic OLF surgery including beak-shaped type and multi vertebral levels. However, whole spine evaluation is important to avoid missing other combined stenoses that may affect outcomes.

## 1. Introduction

Ossification of the ligamentous flavum (OLF) is a relatively common cause of progressive thoracic myelopathy in middle-aged and older men in East Asian countries [1,2]. Although the natural course and prognosis of OLF are not yet known, it has been reported to occur in the lower thoracic spine [3,4,5], near to the conus medullaris. As spinal cord compression progresses, symptoms such as numbness and weakness in the lower extremities, gait disturbance, and bladder-bowel disturbance are expected to progress as well.

Since the duration and severity of preoperative symptoms have been shown to be important prognostic factors [3,6], surgical treatment is strongly recommended for patients with myelopathy [4,6]. Posterior decompression surgery is generally performed [2], which may be supplemented with spinal instrumentation in some cases, with or without the use of microscopy, depending on the institution. While the goal of OLF surgery is to safely and adequately release spinal cord compression, it is also important to minimize dural tearing [7,8] and preserve posterior elements. The spinous process, supraspinatus, and interspinous ligaments are important to prevent kyphosis deformities [9] that can occur postoperatively.

Microendoscopic minimally invasive spine surgery (MISS) of the lumbar spine has been reported to be less invasive than traditional open surgery [10]; therefore, we applied this technique when performing thoracic spine surgery. We previously reported that microendoscopic posterior decompression was effective in nine patients with thoracic OLF under intraoperative neurophysiological monitoring, involvement of a single vertebral level in the lower thoracic area (T9–12), and unilateral or bilateral type without comma and tram track signs as seen on computed tomography (CT) and round morphology as seen on T2-weighted sagittal magnetic resonance imaging (MRI). In this study, a barely adherent part of the ligamentum flavum (LF) was left in situ (to avoid dural tears); however, as much as possible of the surrounding LF was removed (partial floating method).

We continued to perform microendoscopic posterior decompression for thoracic OLF patients and accumulated surgical experience that might expand the surgical indication. In this study, we evaluated characteristic findings of thoracic OLF and the patient’s situation, and consequently determined the latest surgical indication and the factor affected poor operative outcome.

## 2. Materials and Methods

### 2.1. Patient Selection

Microendoscopic posterior decompression was performed on 25 patients with thoracic OLF with myelopathy at Iwai Orthopaedic Medical Hospital, Japan Community Health Care Organization Tokyo Shinjuku Medical Center, and Hangzhou Zhengxing Hospital, from June 2011 to February 2019. Nineteen patients were included in this study; three patients who underwent this surgery simultaneously with lumbar spine surgery and three patients who could not be followed-up for more than three months after surgery were excluded from the study. In total, there were 15 men and four women who were aged between 35 to 81 years (mean age, 61.9 years).

Neurological examination and preoperative T2-weighted MRI were used to identify the location of the OLF and target area for decompression. The changes in signal intensity on MRI of the vertebrae and spinal cord at the corresponding vertebral level were also checked. The morphology of the OLF was classified as unilateral, bilateral, or bridged type by using axial CT images [11], or as round or beak-shaped by using sagittal MRI [11]. The extent of decompression was evaluated by performing pre- and postoperative CT and MRI.

Posterior decompression was performed using the METRx endoscope system (Medtronic Sofamor Danek, Memphis, TN, USA). Patients were followed-up for an average of 17.8 months (range, 5–35 months). Perioperative complications and surgical outcomes were investigated. Pre- and postoperative pains were evaluated using the numerical rating scale (NRS) score. Pre- and postoperative neurological status was evaluated using the modified Japanese Orthopaedic Association (mJOA) score, which has a highest possible score of 11 points from the mJOA score for cervical myelopathy excluding upper extremity function, at the latest follow-up or before additional surgery. The recovery rate was calculated as follows: Recovery rate = postoperative mJOA score − preoperative mJOA score/11 (highest possible score) − preoperative mJOA score × 100

Statistical studies were performed using the corresponding *t*-test and Mann–Whitney *U* test. The threshold for significance was set at *p* < 0.05.

### 2.2. Surgical Technique

Surgery was performed with the patient in the prone position and assisted using intraoperative neurophysiological monitoring for safety. During the operation, a fluoroscope was placed across the center of the operative table to enable appropriate timing. An 18-mm skin incision was made at the target spinal level under fluoroscopic guidance. There are two different approaches to the spinal canal, namely, the midline approach and unilateral paramedian approach [12].

In the midline approach, the tip of spinous processes was exposed, spinous processes were split longitudinally using an electrical high-speed drill (NSK-Nakanishi Japan, Tokyo, Japan), and the paraspinal muscles were spread out using serial tubular dilators with the spinous processes. A 16-mm tubular retractor and endoscope were inserted after exposure of the vertebral lamina. A dome-shaped laminectomy of the corresponding lamina was performed. In the unilateral paramedian approach, the skin and fascial incisions muscle layers were directly split using serial tubular dilators. Hemi-laminectomy was performed after exposure of the approach side of the vertebral lamina, and removal of the inner plate of the contralateral side was performed, subsequently. Whichever approach is used, sufficient laminectomy is required to fully expose the LF.

The first step was to excise the normal ligament and determine the ossified ligament. At this time, the non-fused portion in the midline is an important landmark. The next step was to check the extent of the ossified LF, drill it as thin as possible, carefully detach it from underneath the dura mater, and remove it piece by piece using a small-angled Kerrison rongeur (width: 1 and 2 mm). The tightly adherent region of the ossified LF remained but was completely isolated from the surrounding LF. During the last stage of the operation, we confirmed good pulsatile movement of the remaining LF together with the surrounding dura mater. This technique is called the partial floating method [12]. The decompression was completed after confirmation of the improvement in the amplitude of motor evoked potentials in at least one muscle area. Furthermore, the extent of decompression was confirmed using fluoroscopy. After decompression, the working channel was carefully removed, and a drain was placed. The fascia and skin were closed using standard techniques.

The patients were allowed to get up from the bed with a thoracolumbar spine brace the day after surgery. The drain was removed two days after surgery. Patients were from hospital approximately one week after surgery; however, the length of hospital stay was often affected by perioperative neurological status.

### 2.3. Ethical Standards

All the procedures performed in studies involving human participants were in accordance with the ethical standards of the research committee of Iwai Medical Foundation (No. 20110117, 17 January 2011) and with the 1964 Helsinki Declaration. Informed consent was obtained by the disclaimer documents for the surgical procedure handed over to the patient with explanations and signed.

## 3. Results

### 3.1. Characteristic Features and Result

Eighteen patients underwent surgery at one vertebral level and one patient underwent surgery at two vertebral levels. All OLFs were found in the lower thoracic vertebrae from T9 to T12 (T9/10 in two patients, T10/11 in eight patients, and T11/12 in 10 patients). Axial CT showed all cases were of the bilateral type, except for one (Patient 14), which was unilateral; sagittal MRI showed that all cases had round morphology, except for one (Patient 15), which was beak-shaped. The tram track sign is a hyperdense bony excrescence with a hypodense center, and the comma sign is evidence of ossification of one-half of the circumference of the dura mater revealed on CT. Neither sign, both of which are known to be predictive of dural adhesion [2,13,14], was observed in any patient (Table 1).

### 3.2. Perioperative Complications

Limited laminectomy and flavectomy were performed under microendoscopic visualization. Fifteen patients underwent the unilateral paramedian approach (Figure 1), and four patients underwent a midline approach (Figure 2). The partial floating method was performed in six cases because the OLF adhered to the dura mater. The average operative time per vertebrate was 99 min (range, 62–170 min), with negligible blood loss in all cases and no cases requiring blood transfusion. 

A small dural tear occurred in one patient (Patient 5); however, no cerebrospinal fluid leakage was observed, and direct repair was not necessary. Measurement of Motor Evoked Potential (MEP) was performed in 17 patients to avoid intraoperative spinal injuries. Abnormal changes in MEP (a decrease of more than 50% in the amplitude or a 2 ms delay in the latency) were observed in one patient (Patient 3), in whom a temporary decrease in the amplitude was seen during the manipulation of the ossified LF. Surgical manipulation was discontinued immediately and only resumed when the MEP amplitude had recovered. One patient (Patient 11, Figure 2) developed bilateral lower extremity paralysis due to postoperative hematoma that resulted in emergency reoperation on the day of surgery. The patient underwent an initial surgery at the level of T11/12 (Patient 10, Figure 3), but due to worsening of the T10/11 stenosis in the postoperative course and poor improvement in waddling gait and lower extremity numbness, additional endoscopic decompression was performed.

### 3.3. Surgical Outcome and Combined Spinal Lesions in Patients with OLF

The mean pre- and postoperative NRS scores in all cases were 6.6 and 5.3, respectively. The mean postoperative NRS score improved statistically (*p* < 0.01). In addition to thoracic spinal stenosis, 12 patients had symptomatic cervical and/or lumbar spine stenosis. In nine cases, other surgeries were required before or after thoracic spine surgery. Eight cases had a history of cervical or lumbar spine surgery prior to thoracic spine surgery; four required additional cervical or lumbar spine surgery after thoracic spine surgery. We therefore reanalyzed the mean NRS scores in two different subgroups (the patients who underwent thoracic spine surgery alone and the patients who underwent other cervical or lumbar spine surgery). There was a statistically difference between the mean NRS scores in the former group (*p* < 0.01) but not in the latter group (*p* = 0.06) (Figure 4).

We also evaluated the surgical outcome using the mJOA score. Twelve patients showed an increase in the mJOA score, while only one patient showed a decline at the postoperative follow-up because of postoperative hematoma; the other six patients remained unchanged. The average pre- and postoperative mJOA scores were 7.2 and 8.4, respectively, and the mean recovery rate was 33% at a mean follow-up of 17.9 months (range, 6–35 months). There was no significant difference between midline (18.8%) and lateral paramedian (36.9%) approaches regarding the rate of improvement in mJOA scores; however, the lateral paramedian approach tended to have slightly better improvement (*p* > 0.05). There was a significant difference in the recovery rates between patients who underwent thoracic spine surgery alone (50.5%) and those who underwent cervical or lumbar spine surgery (13.7%) (0.01 < *p* < 0.05) (Table 2). 

## 4. Discussion

OLF is a known cause of thoracic myelopathy, and surgery is preferable when spinal myelopathy develops. While there have been various reports on different surgical methods [11,12,13,14,15,16,17,18,19,20,21,22,23], there are no conclusions yet as to which surgical procedure is associated with best outcomes. The microendoscopic surgery technique has gradually been developed and become more popular for the lumbar spine [24,25,26]. More recently, it has been applied to the cervical [27,28] and thoracic spine [29] areas, and only a few case reports are available on the use of this approach for the treatment of thoracic OLF. Ikuta et al. reported successful microendoscopic decompression of the OLF using a midline approach [30], commonly called the muscle-preserving interlaminar decompression (MILD) method, in the lumbar spine. Although they reported the usefulness of endoscopic surgery, they concluded that because of technical difficulties, this procedure is only indicated for particular cases of thoracic OLF, such as those without fusion in the middle of the spinal canal and OLF in the absence of dural ossification.

We also reported the experiences of endoscopic posterior decompression for thoracic OLF [12] and proposed the following surgical indication from the 9 cases experiences: (1) a single vertebral level, (2) in a lower thoracic area (T9–12), (3) unilateral or bilateral type without comma or tram track signs on CT, (4) round morphology as seen on T2-weighted sagittal MRI, (5) barely adherent part of the LF left in situ (to avoid dural tears; however, as much of the surrounding LF should be removed as much as possible (partial floating method)); and (6) intraoperative neurophysiological monitoring of sequential MEPs.

Since then, ten more patients were operated on, and we have had new experiences. The tram track sign and the comma signs, which are known to be predictive of dural adhesion, were not observed in any patient; there was only one case (Patient 3) of dural tear. There is one case each of the morphology classified as unilateral type (Patient 14) using axial CT and beak-shaped (Patient 15) using the sagittal slice of MRI. Unilateral type was a very good indication for unilateral approach, and the operative time was shorter (68 min) than the average (99 min). On the other hand, the beak-shaped type required more careful manipulation, and the operative time was longer (117 min) than the average (99 min). There was also one case (Patient 16) of surgery at two adjacent vertebral levels at the same time. This took approximately twice as long (170 min) as surgery was performed at one vertebral level; however, it has been found to be applicable to beak-shaped and multiple vertebral levels. We therefore propose to include following 2 criteria in new surgical indication: (1) multiple vertebral levels; (2) beak-shaped morphology as seen on T2-weighted sagittal MRI. 

We also encountered two cases that resulted in additional postoperative surgery. In one case, postoperative hematoma caused complete paralysis, and emergency surgery was performed on the day of surgery (Patient 11, Figure 2). During the operation, the patient was normal, but during extubation, he became uncomfortable and stood up on the stretcher. A sudden increase in blood pressure was the likely cause of bleeding. Reoperations were performed using the same approach with endoscopy to remove the hematoma and stop the bleeding from the epidural space and muscle layers. In addition to the usual hemostatic maneuvering, Furoseal^®^ (Baxter Limited, Chicago, IL, USA) was also used. The patient continued inpatient rehabilitation and recovered to near his original level of activities of daily living by the final follow-up, which took nearly six months. Microendoscopic surgery is known to have a small dead space; therefore, postoperative hematoma is more likely to lead to complications such as paralysis. Postoperative hematoma, which is a greater risk with this microendoscopic surgery, may be difficult to prevent completely, “therefore, attention should be paid to detect its formation. In another case, six months postoperatively, adjacent thoracic canal stenosis occurred requiring reoperation (Patient 10, Figure 3). The patient underwent an initial surgery at the level of T11/12, but due to worsening of T10/11 stenosis postoperatively and poor improvement in gait instability and lower extremity numbness, additional endoscopic decompression was performed. The adjacent region had a small OLF; however, MRI showed no spinal intensity change, and this area was not considered for surgery initially. Sagittal CT after the initial surgery showed that the T10 spinous process was nearly resected, which may have exacerbated the instability of the adjacent T10/11.

Before our experiences with thoracic OLF treated by conventional open surgery, there were only 7 cases (Appendix A), and the surgical indication is different from that of microendoscopic posterior decompression as follows: (1) bridged type, (2) diffuse idiopathic skeletal hyperostosis (DISH) at adjacent vertebral bodies, (3) upper thoracic vertebral level, (4) the case that combined operation of plural spinal areas should be performed at one time. Therefore, it is difficult to compare the surgical outcome without bias of the background. The average pre- and postoperative mJOA scores were 6.3 and 6.4, respectively, in our open surgery group. This bad outcome might indicate that our open surgery group included more severe thoracic OLF cases. In regards to the operative time per one vertebral level, the average time was shorter (90 min) than that of microendoscopic group (99 min). This indicates a high degree of difficulty in microendoscopic technic. In terms of complications, one surgical site infection (SSI) occurred in the open surgery group. Previously 3.6–5.8% of SSI had been reported for thoracic OLF treated by conventional open surgery [31,32]. As we did not experience SSI in 19 cases of microendoscopic group, SSI may be suppressed by microendoscopic decompression.

More recently, some reports of full endoscopic spine surgery for thoracic OLF have resulted in good postoperative outcomes with less invasion [33,34,35]. Our group also reported the benefits of full endoscopic surgery for OLF of the lumbar spine [36]; however, these benefits are somewhat doubtful for OLF of the thoracic spine. This is because we are concerned about the risk of iatrogenic spinal cord injury due to unstable retractor use in full endoscopic surgery at the thoracic level and the unknown pressure effect of physiologic saline irrigation. 

Spinal stenosis at other sites is commonly observed in thoracic OLF patients; therefore, it is important to evaluate other lesions outside the surgical site. Alternatively, 12 of the 19 patients were affected at other sites, and nine patients had a history of surgery before or after thoracic spine surgery. Whole spine evaluation is desirable to improve the patient’s prognosis. 

Several case series have been reported regarding the open surgical outcome of thoracic OLF. Most studies assessed the outcome based on the recovery rate of the mJOA score, ranging 16–58.7% [11,13,14,15,16,17,18,19,20,21,22,23]. In our case, the mean recovery rate of 33% after a mean follow-up of 18.2 months was lower than expected. However, the recovery rate increased to 50.5%, only analyzing the cases without other spinal disorders. There is still controversy on whether other coexisting spinal disorders are predictive of poor outcome [37]; at least surgery in other spinal areas was predictive of poor outcome for our microendoscopic decompression. 

## 5. Conclusions

Preliminary results in this small number of patients show that microendoscopic posterior decompression involving a thoracic laminectomy combined with MEP monitoring is feasible for the treatment of patients with limited type of thoracic OLF that are free of both the tram track sign and comma sign, which are known to predict dural adhesions. It is important to evaluate the entire spine because long-term surgical outcomes are significantly worse in cases with a history of cervical or lumbar spine surgery before or after thoracic spine surgery compared to that of patients who underwent thoracic spine surgery alone.

## Figures and Tables

**Figure 1 medicina-56-00684-f001:**
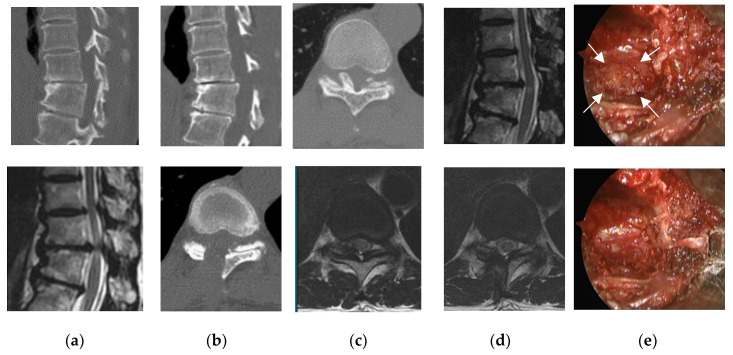
Pre- and postoperative findings and views of typical unilateral approach (Patient 16). Computed tomography scan, (**a**) preoperative, (**b**) postoperative; T2-weighted magnetic resonance imaging, (**c**) preoperative, (**d**) postoperative. (**e**) intraoperative photographs: after removal of the ligamentum flavum (LF) on the right side (top image), the right side of the ossified LF (arrows) was completely removed (bottom image).

**Figure 2 medicina-56-00684-f002:**
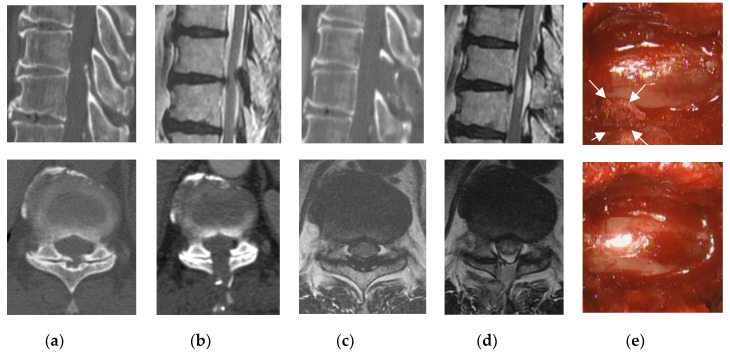
Pre- and postoperative findings and views of typical midline approach (Patient 11). Computed tomography scan, (**a**) preoperative, (**b**) postoperative; T2-weighted magnetic resonance imaging, (**c**) preoperative, (**d**) postoperative. (**e**) Intraoperative photographs: After removal of the ligamentum flavum (LF) on the left side (top image), the left side of the ossified LF (arrows) was completely removed (bottom image).

**Figure 3 medicina-56-00684-f003:**
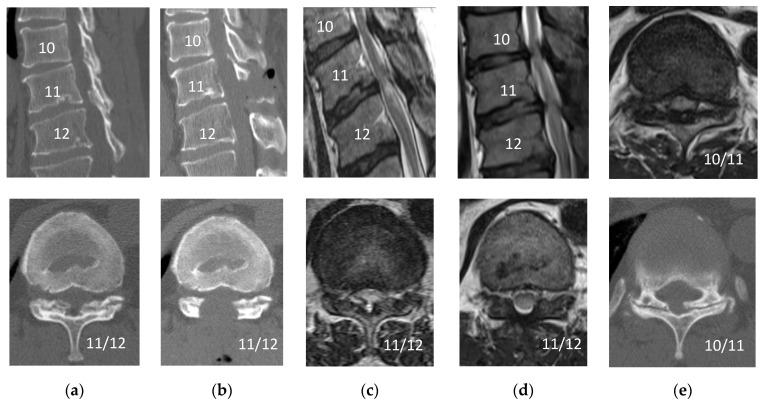
Pre- and postoperative findings (Patient 10). Computed tomography scan, (**a**) preoperative (T11/12), (**b**) postoperative (T11/12), (**e**) preoperative T10/11; T2-weighted magnetic resonance imaging, (**c**) preoperative (T11/12), (**d**) postoperative (T11/12), (**e**) postoperative (T10/11): The patient underwent an initial surgery at the level of T11/12, the T10/11 stenosis has worsened postoperatively.

**Figure 4 medicina-56-00684-f004:**
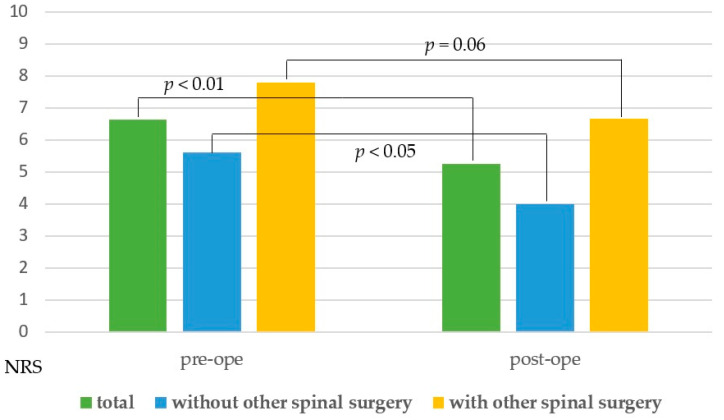
Bar chart for the mean pre- and postoperative NRS scores. NRS = numerical rating scale; Pre-ope = preoperative; Post-ope = postoperative.

**Table 1 medicina-56-00684-t001:** Summary of characteristic features and result.

Patient	Age	Sex	Level	Type (Axial CT)	Type (Sagittal MRI)	Intensity Change (MRI)	Approach	Operation Time (minutes)	Resection
1	64	M	T11/12	bilateral	round	−	unilateral	166	Partial
2	67	M	T11/12	bilateral	round	+	midline	112	Partial
3	35	M	T10/11	bilateral	round	+	unilateral	101	Complete
4	70	F	T10/11	bilateral	round	−	unilateral	85	Partial
5	62	M	T10/11	bilateral	round	+	unilateral	87	Complete
6	62	M	T11/12	bilateral	round	+	unilateral	112	Partial
7	76	M	T9/10	bilateral	round	+	unilateral	62	Complete
8	64	F	T10/11	bilateral	round	−	midline	112	Partial
9	38	M	T11/12	bilateral	round	+	unilateral	140	Complete
10	55	M	T11/12	bilateral	round	+	midline	118	Partial
11	81	M	T10/11	bilateral	round	−	midline	76	Complete
12	74	M	T9/10	bilateral	round	+	unilateral	96	Complete
13	41	M	T11/12	bilateral	round	+	unilateral	106	Complete
14	76	M	T11/12	unilateral	round	−	unilateral	68	Complete
15	34	M	T10/11	bilateral	beak	+	unilateral	117	Complete
16	56	F	T10/11/12	bilateral	round	+	unilateral	170	Complete
17	79	M	T11/12	bilateral	round	+	unilateral	75	Complete
18	73	F	T10/11	bilateral	round	+	unilateral	68	Complete
19	70	M	T11/12	bilateral	round	+	unilateral	106	Complete

CT: computed tomography, MRI: magnetic resonance image, M: male, F: female, −: no intensity change is present, +: intensity change is present.

**Table 2 medicina-56-00684-t002:** Results, perioperative complications, and combined spinal lesions in patients with OLF.

Patient	Pre-mJOA	Post-mJOA	Recovery Rate (%)	Follow Up Month	Complication	Combined Spinal Lesions
1	7	10	75	25	−	C+ pre/L+ post
2	5	6	17	13	−	N
3	9	11	100	7	−	C−
4	2	4	22	25	−	L−
5	6	8	40	23	dural tear	C+ post/L+ pre
6	7	7	0	36	−	N
7	9	9	0	25	−	L+ post
8	9	10	50	24	−	C−
9	8	11	100	7	−	N
10	8	9	33	6	re-operation for adjacent level stenosis	N
11	7	6	−25	29	re-operation for paralysis due to hematoma	C+ post/L+ post
12	7	7	0	5	−	C+ post/L+ pre
13	8	10	67	16	−	N
14	9	9	0	10	−	L+ pre
15	5	10	83	11	−	N
16	8	9	33	22	−	N
17	7	7	0	19	−	L+ post
18	8	9	33	19	−	L+ post
19	7	7	0	17	−	L+ post

OLF: ossification of the ligamentum flavum, mJOA: modified Japanese Orthopedic Association, −: no complication, C+: cervical canal stenosis for operation, C−: cervical canal stenosis for observation, L+: lumber canal stenosis for operation, L−: lumber canal stenosis for observation, N: no operation, post means operation before the thoracic OLF, pre means operation after the thoracic OLF.

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
