# Peer review of "Microendoscopic Posterior Decompression for Treating Thoracic Myelopathy Caused by Ossification of the Ligamentum Flavum: Case Series"

_medicina, 2020, doi:10.3390/medicina56120684_

Round 1

Reviewer 1 Report

I read your article with interest, there are only a limited amount of papers that discuss thoracic myelopathy due to OLF, and there I enjoyed reading your paper. The main concern I have with the paper is the choice of mJOA as an outcome measure. This outcome measure was designed for cervical myelopathy and as such contains as part of its evaluation upper limbs assessment components. You have removed this part out of the score which was correct but which leaves your outcome assessment very limited. It would have been great to see how patients were in terms of quality of life, had pain, or other general outcomes as well. 

I would suggest that you also focus your paper a little better. Are you presenting this research to show the positive outcome of surgical intervention? It is clear that surgery for this condition, as with cervical myelopathy, is undertaken to decompress the spinal cord regardless of method. If you which to highlight the positives of one approach over another, I would discuss complication rates, operative time, radiological outcomes between the endoscopic methods vs other methods.

Reviewer 2 Report

I appreciate having the opportunity to review your valuable research.

After reviewing this paper, I found several major problems in ‘Purpose and Methodology’ as followings. If these problems are not resolved, it is difficult to be accepted as a meaningful clinical paper.

1. What is the purpose of this paper? If you are investigating the effects of minimally invasive surges for thoracic OPLL, you need to focus it in ‘Conclusion’.

2. The number of participants in this paper is very small to identify the effect of minimally invasive surges for thoracic OPLL. In order to investigate the purpose of this study, the superiority of minimally invasive surges should be demonstrated against to conventional surgery under the comparative study design. Unfortunately, in this study, there is no these scientific process.

3. If you want to share your experiences with the readers, I recommend you to rewrite it as ‘Case Series’.

Sincerely,

Round 2

Reviewer 1 Report

The manuscript is improved, thanks. There are however minor improvements in language necessary throughout the manuscript.

Reviewer 2 Report

 This paper well described rare surgical experiences as a form of case series. In the future, I expect great results as a form of clinical article with large number of patients. The authors' experiences have been described in detail. In addition, I think the points recommended in 1st review have been also fixed sufficiently.

Thank you for your efforts. I personally had a good experience as a reader and reviewer.